# GDP-Mannose Pyrophosphorylase B (*GMPPB*)-Related Disorders

**DOI:** 10.3390/genes14020372

**Published:** 2023-01-31

**Authors:** Pitcha Chompoopong, Margherita Milone

**Affiliations:** Department of Neurology, Division of Neuromuscular Medicine, Mayo Clinic, Rochester, MN 55905, USA

**Keywords:** GMPPB, dystroglycanopathy, limb-girdle muscular dystrophy, congenital muscular dystrophy, congenital myasthenic syndrome

## Abstract

GDP-mannose pyrophosphorylase B (GMPPB) is a cytoplasmic protein that catalyzes the formation of GDP-mannose. Impaired GMPPB function reduces the amount of GDP-mannose available for the O-mannosylation of α-dystroglycan (α-DG) and ultimately leads to disruptions of the link between α-DG and extracellular proteins, hence dystroglycanopathy. GMPPB-related disorders are inherited in an autosomal recessive manner and caused by mutations in either a homozygous or compound heterozygous state. The clinical spectrum of GMPPB-related disorders spans from severe congenital muscular dystrophy (CMD) with brain and eye abnormalities to mild forms of limb-girdle muscular dystrophy (LGMD) to recurrent rhabdomyolysis without overt muscle weakness. *GMPPB* mutations can also lead to the defect of neuromuscular transmission and congenital myasthenic syndrome due to altered glycosylation of the acetylcholine receptor subunits and other synaptic proteins. Such impairment of neuromuscular transmission is a unique feature of GMPPB-related disorders among dystroglycanopathies. LGMD is the most common phenotypic presentation, characterized by predominant proximal weakness involving lower more than upper limbs. Facial, ocular, bulbar, and respiratory muscles are largely spared. Some patients demonstrate fluctuating fatigable weakness suggesting neuromuscular junction involvement. Patients with CMD phenotype often also have structural brain defects, intellectual disability, epilepsy, and ophthalmic abnormalities. Creatine kinase levels are typically elevated, ranging from 2 to >50 times the upper limit of normal. Involvement of the neuromuscular junction is demonstrated by the decrement in the compound muscle action potential amplitude on low-frequency (2–3 Hz) repetitive nerve stimulation in proximal muscles but not in facial muscles. Muscle biopsies typically show myopathic changes with variable degrees of reduced α-DG expression. Higher mobility of β-DG on Western blotting represents a specific feature of GMPPB-related disorders, distinguishing it from other α-dystroglycanopathies. Patients with clinical and electrophysiologic features of neuromuscular transmission defect can respond to acetylcholinesterase inhibitors alone or combined with 3,4 diaminopyridine or salbutamol.

## 1. Introduction

Dystroglycanopathies are a group of clinically and genetically heterogeneous muscular dystrophies caused by mutations that lead to a defect in α-dystroglycan (α-DG) [1]. Primary dystroglycanopathy is a very rare limb-girdle muscular dystrophy caused by a mutation in the *DAG1* gene, which encodes the dystrophin-associated glycoprotein 1 [2]. Most dystroglycanopathies are due to mutations in genes encoding proteins with enzymatic function involved in the glycosylation of α-DG and are secondary dystroglycanopathies. Eighteen proteins are known to cause secondary dystroglycanopathies by altering α-DG glycosylation, and GDP-mannose pyrophosphorylase B (GMPPB) is one of these proteins [3]. Glycosylation is a post-translational protein modification process in which sugar moieties are added to proteins as they pass through the endoplasmic reticulum and Golgi apparatus. Glycosylation of α-DG is essential for its interactions with extracellular proteins [4,5]. GMPPB catalyzes the formation of guanosine diphosphate (GDP) mannose, which is a key substrate for multiple glycosylation pathways, including O-mannosylation of α-DG. Hence, mutations in *GMPPB* result in secondary dystroglycanopathy. Like other dystroglycanopathies, *GMPPB* mutations manifest with a spectrum of clinical features of variable severity, spanning from congenital muscular dystrophies (CMD) with brain and eye abnormalities to limb-girdle muscular dystrophies (LGMD) to exercise intolerance and rhabdomyolysis [6,7]. Differently from other dystroglycanopathies, *GMPPB* mutations can lead to a defect of neuromuscular transmission and congenital myasthenic syndrome due to altered glycosylation of the acetylcholine receptor subunits and other synaptic proteins [8,9,10]. This unique feature is clinically important as the majority of patients with impaired neuromuscular transmission can experience a meaningful improvement in strength and endurance with cholinergic drugs.

This review article aims to summarize the clinical and molecular spectrum of GMPPB-related disorders, emphasizing the unique involvement of muscle and neuromuscular junction, which has therapeutic implications relevant to patient care.

## 2. GDP-Mannose Pyrophosphorylase B (GMPPB) and Its Association with α-Dystroglycan (α-DG)

GMPPB is a cytoplasmic protein that catalyzes the formation of GDP-mannose from GTP and mannose-1-phosphate (Figure 1). GDP-mannose is the substrate of cytosolic mannosyltransferases required for the synthesis of dolichol-phosphate mannose (Dol-P-Man). This is the essential mannose donor for mannosylation reactions in the endoplasmic reticulum, including O-mannosylation, C-mannosylation, N-glycosylation, and glycosylphosphatidylinositol anchor formation [6,11]. Human GMPPB has two main functional domains: the N-terminal nucleotidyl transferase domain (harboring the conserved signature motif for nucleotide binding and transfer) and the C-terminal bacterial transferase hexapeptide domain (forming a left-handed β-helix structure).

α-DG is an extracellular component of the dystroglycan complex. It is non-covalently linked to the transmembrane β-DG and together provides a link between the extracellular matrix and intracellular machinery [12]. α-DG is a heavily glycosylated protein, and its interaction with extracellular proteins, such as laminin-2, agrin, perlecan, neurexin, nidogen, pikachurin, and slit, is mediated by its sugar components [13]. Most glycans attached to α-DG are O-linked via the sugar mannose, O-mannosylation [4]. Impaired GMPPB function reduces the amount of GDP-mannose available for O-mannosylation of α-DG and ultimately leads to disruptions of the link between α-DG and extracellular proteins. Similarly, mutations in genes encoding for O-mannosyltransferase (*POMT1, POMT2, PMGNT1*) [14,15,16] and proteins involved in the synthesis of Dol-P-Man (*DPM2, DPM3*, and *DOLK*) [17,18,19] also disrupt α-DG O-mannosylation and cause dystroglycanopathies.

## 3. Gene and Mutations 

The *GMPPB* gene is mapped to chromosome 3p21.31 and encodes the β subunit of the GMPPB enzyme. In humans, *GMPPB* is transcribed as two isoforms: a shorter isoform consisting of 8 coding exons and 387 amino acids and a longer isoform consisting of 9 coding exons and 360 amino acids. The longer isoform is expressed in skeletal muscle (fetal and adult) and brain at higher levels than the shorter isoform [6]. GMPPB-related disorders are inherited in an autosomal recessive manner and caused by mutations in either a homozygous or compound heterozygous state [6]. To date, more than 50 pathogenic mutations in *GMPPB* have been reported. They reside in either the N-terminal or C-terminal of the protein or the inter-domains (Table 1). Most mutations are missense mutations, with two being the most common, c.79G>C (p.Asp27His) and c.860G>A (p.Arg287Gln). Up to date, more than half of the reported individuals with GMPPB-related disorder are compound heterozygous for one of these two mutations [20,21,22]. Additional proposed protein hotspots include residues Pro32, Pro103, and Arg185 [21].

## 4. Prevalence

The frequency of *GMPPB* mutations is estimated to be approximately 5% among patients with dystroglycanopathies, based on a large cohort study from China [32]. Based on clinical phenotypes and according to studies on genetically characterized large cohorts of patients in the United Kingdom, Italy, and China, pathogenic *GMPPB* mutations are responsible for 2–3% of patients with CMD and 5–7% of patients with LGMD phenotype [32,41,42]. *GMPPB* mutations cause approximately 5–7% of the genetically proven CMS patients in Spain, China, and Austria [23,34,40]. The frequency of the co-existing involvement of the muscle and neuromuscular junction in the same patient is unknown and may be underestimated. Indeed, the diagnosis of muscular dystrophy in a patient with LGMD phenotype and elevated creatine kinase (CK) levels is not always complemented by repetitive nerve stimulation to look for associated neuromuscular transmission defect. Moreover, the defect of neuromuscular transmission may be undetected if repetitive nerve stimulation is performed in an unaffected facial or distal muscle.

Conversely, the diagnosis of CMS is not always accompanied by a muscle biopsy to search for evidence of myopathy through histological and immunohistochemical studies.

## 5. Clinical Phenotype and Creatine Kinase Levels

The clinical spectrum of *GMPPB*-related disorders spans from severe CMD with brain and eye abnormalities (muscle-eye-brain/Fukuyama congenital muscular dystrophy-like) to milder forms of muscular dystrophy with or without overlapping CMS features to recurrent rhabdomyolysis without overt muscle weakness or asymptomatic hyper-CK-emia [6,7,8,10,20,24]. Among these phenotypes, LGMD (LGMD2T) is the most common occurring in approximately 60–70% of patients with *GMPPB* mutations [20,21].

### 5.1. LGMD

Patients with LGMD phenotype manifest in childhood or adulthood through the fourth decade. The weakness is predominantly proximal, affecting the shoulder and pelvic girdle muscles and involving the lower more than the upper limb muscles [21]. Patients may present with an inability to get up from the floor, climb stairs, or run [10]. Most patients recall poor performance in sports or being slower than their peers in childhood [8,10]. Axial weakness resulting in lumbar hyperlordosis [10] and calf enlargement have been reported [24,25]. Ocular, facial, bulbar, and respiratory muscles are largely spared [6,8,24]. However, respiratory involvement in late adulthood (around age 70) has been described [24]. Some patients experience painful muscle cramps, exercise intolerance, and myoglobinuria in association with the weakness or prior to the onset of weakness; others display myoglobinuria and hyper-CK-emia without overt muscle weakness, mimicking metabolic myopathies [7,20,24]. Cardiac conduction defects have been infrequently reported and include long QT intervals, first-degree AV block, and Wolff–Parkinson–White syndrome [7,20,24]. Sudden heart block has been also documented but in a patient with congenital disease [20]. Cognitive functions are typically spared, although mild intellectual disability and epilepsy have been observed [6,20]. Muscle weakness is usually progressive over time; intermittent exacerbations (especially after intercurrent illnesses) may occur in some patients [10,26,38]. Patients with adulthood onset are more frequently found to have evidence of neuromuscular junction involvement [8,20].

### 5.2. CMD

Patients with CMD phenotype are more severely affected. They present at birth with hypotonia, generalized or proximal and axial weakness, motor developmental delay, and may show respiratory distress at birth or within the first few months of life [6,20]. Prenatal presentation with abnormal intrauterine growth and decreased fetal movements can occur [22]. Some patients with CMD phenotype can achieve the ability to ambulate, while others are unable to sit without support [20]. Arthrogryposis, congenital clubfoot, and scoliosis can develop in patients with early disease onset. Scoliosis resulting in the loss of ambulation and requiring surgery can occur [22]. In addition to skeletal muscle weakness, patients with CMD phenotype may have ophthalmic involvement (cataract, strabismus, and nystagmus), epilepsy, and microcephaly, similar to what is observed in other dystroglycanopathies [6,20]. Most patients with congenital onset show intellectual disability, which typically affects the language domain. Its severity can range from the inability to produce full sentences to autism spectrum disorder [20]. Intellectual disability is common and can be a predominant feature in CMD-GMPPB, but it has also been reported in adult patients, especially in those with skeletal myopathy manifesting before age 18 [10,20,21]. Epilepsy manifests with generalized tonic–clonic or focal seizures with impaired awareness and oromasticatory automatisms [20]. An MRI of the brain may show structural defects, such as polymicrogyria or cortical hypoplasia, microcephaly, white and gray matter changes, and ventriculomegaly. Cerebellar hypoplasia was reported in 30% of patients [6,24]. In a study, epilepsy did not correlate with brain MRI abnormalities [20].

### 5.3. CMS

Mutations in *GMPPB* can lead to neuromuscular transmission defects. In such cases, patients show a clinical phenotype featured by fluctuating fatigable muscle weakness [8,10]. They may manifest muscle fatigability in the setting of fixed weakness or without fixed overt weakness, featuring a classic myasthenic syndrome. Compared to other CMS, patients with GMPPB-CMS have a more frequent later age of onset in adolescence or adulthood, although, retrospectively, many patients report difficulty keeping up with peers in childhood or being poorly athletic. GMPPB-CMS usually spares ocular, facial, bulbar, and respiratory muscles. The degree of muscle fatigability can be quite dramatic to the point that patients may fluctuate from being able to ambulate for short distances with an aid to being unable to stand independently [8]. Exacerbation of muscle weakness can sometimes be triggered by viral infections or menstruation, as also observed in other myasthenic disorders [10]. Clinical examination may not be sufficiently sensitive to easily demonstrate abnormal muscle fatigability, especially in the setting of baseline fixed weakness. Hence, electrodiagnostic studies are often required to identify the defect of neuromuscular transmission [10]. Additional laboratory features, such as the elevated CK values and dystrophic changes on muscle biopsy, can help in differentiating the GMPPB-related neuromuscular junction defect from other CMS. It is important to note that not all patients with *GMPPB* mutations have electrophysiological evidence of a defect of neuromuscular transmission; however, if present, it has therapeutic implications.

### 5.4. Creatine Kinase (CK) Levels

Regardless of the clinical phenotype, CK levels at baseline are typically elevated, ranging from 2 to >50 times the upper limit of normal [7,10,27].

## 6. Electrophysiologic Findings

Significant decrement in the compound muscle action potential amplitude (CMAP) on low-frequency (2–3 Hz) repetitive nerve stimulation is present in many patients with GMPPB-related disorders (Figure 2) [8,10]. An increment in CMAP amplitude (up to 70%) following maximal volitional contraction or high-frequency nerve stimulation has also been reported [10]. The defect of neuromuscular transmission is usually present in proximal muscles, in keeping with the patient’s pattern of muscle weakness, while facial muscles are spared not only by repetitive nerve stimulation but also by single-fiber (SF) electromyography (EMG) studies. Repetitive compound muscle action potential (R-CMAP) has not been observed in any patient. In vitro microelectrode studies in a single patient demonstrated impairment of both post-synaptic and pre-synaptic sites of the neuromuscular junction [9]. Miniature endplate potentials had reduced amplitude (about 60% of normal), in keeping with the reduced number of acetylcholine receptors at the endplates, and mildly prolonged decay time, supportive of a post-synaptic defect of neuromuscular transmission [9]. This post-synaptic defect is thought to be due to impaired glycosylation of the acetylcholine receptor (AChR) subunits and can be observed not only in patients with *GMPPB* mutations but also in association with mutations in genes encoding other proteins with a crucial role in glycosylation (i.e., *ALG2, ALG14, DPAGT1, and GFPT1*), also leading to congenital myasthenic syndrome [10]. N-glycosylation of the AChR subunits is indeed essential for the proper folding of their functionally relevant domains, assembly into a pentameric structure, and efficient expression on the surface of post-synaptic folds. Thus, a defect in glycosylation resulting in impaired glycosylation of the AChR subunit results in a decreased number of AChRs at the neuromuscular junctions [43]. Pre-synaptic dysfunction was proven by the reduced quantal content of endplate potentials (about 40% of normal) due to a low number of releasable quanta, while the probability of quantal release was spared. It was suggested that impaired agrin glycosylation could be responsible for the facilitation, as agrin mutations can lead to facilitation, and agrin decreased glycosylation has been demonstrated in a mouse model of agrin-CMS [44,45].

Needle EMG of the affected muscles frequently shows low amplitude, short duration polyphasic motor unit potentials with early recruitment suggestive of myopathy in most patients. Normal EMG findings have been reported in some patients [20]. A report described abnormal spontaneous muscle activity in a patient with GMPPB muscular dystrophy [28]. Motor unit potential variability occurs in the presence of the neuromuscular transmission defect [9]. Peripheral neuropathy has not been reported in patients with a GMPPB disorder.

## 7. Muscle Biopsy Findings

Muscle biopsies typically show myopathic changes with variable degrees of reduced α-DG expression (Figure 3 and Figure 4) [6,24]. The spectrum of histological changes varies from mild nonspecific myopathic changes to frank dystrophic features. Central nucleation is common and can occasionally feature centronuclear myopathy [9,24]. Minimal interstitial inflammatory reaction has been occasionally reported [24]. Tubular aggregates, occurring in some patients with CMS due to mutations in genes involved in glycosylation pathways, have not been observed in *GMPPB* disorder patients.

The level of α-DG glycosylation can be assessed by immunohistochemistry using antibodies recognizing the glyco-epitopes, such as IIH6 and VIA4-1. The IIH6 epitope specifically recognizes the G-domain binding site of α-DG [12,21]. Muscle biopsy of patients with GMPPB-related disorders, immunostained with antibodies directed against the IIH6 epitope, shows variable reduction of α-DG expression, while α-DG expression appears normal when the muscle is immunostained with antibodies directed against the core protein [6]. β-DG immunoreactivity is normal. Secondary reduction of laminin-α2 immunoreactivity occurs in some but not all patients [21].

Western blotting of muscle biopsies from patients with GMPPB-related disorders confirms the reduction in glycosylated α-DG in most patients [20,21,24]. Occasional patients have shown no α-DG reduction in adulthood (but patchy α-DG expression by immunostaining) or early childhood, but α-DG reduction was evident on follow-up immunoblotting a few years later [24]. Higher mobility of β-DG on Western blotting (reflected as a slightly lower molecular mass) has also been consistently observed in the tested muscle biopsies of patients with *GMPPB*-related muscular dystrophy but not in patients with α-dystroglycanopathies caused by mutations in other genes. Thus, the β-DG electrophoretic mobility change was suggested to represent a specific feature of GMPPB-related disorders, distinguishing it from other α-dystroglycanopathies. It was speculated that the β-DG mobility shift in patients with *GMPPB* mutations could be due to its abnormal N-linked glycosylation, as GMPPB is crucial for both O-linked and N-linked glycosylation [21].

Ultrastructural studies of the neuromuscular junction in a patient with GMPPB disorder featuring a defect of neuromuscular transmission showed simplified, absent, and degenerating junctional folds, dilated vesicles in the junctional sarcoplasm, and structurally normal pre-synaptic nerve terminals [9].

## 8. Muscle MRI Findings

Muscle MRI has shown selective fibrous and fatty changes in lower limb muscles, with predominant involvement of posterior and proximal muscles [8], in keeping with the clinical phenotype. Relative sparing of vastus intermedius, sartorius, and gracilis muscles has been reported. Some patients have shown preferential or selective involvement of paraspinal muscles [21,25]. Some young pediatric patients have demonstrated no specific pattern of muscle involvement [20].

## 9. In Vitro Muscle Analysis

Wild-type GMPPB is a soluble cytoplasmic enzyme. In vitro studies of *GMPPB* mutants can lead to an increased propensity for protein aggregation and altered subcellular localization. The p.Pro32Leu, p.Val254Met, p.Arg287Gln, p.Asp334Asn, and p.Arg357His mutations cause GMPPB aggregation within the cytoplasm; p.Pro22Ser cause the protein to aggregate near membrane protrusions into the cytoplasm; p.Asp27His and p.Arg185Cys do not cause discernible changes, and the mutant GMPPB remain evenly distributed throughout the cytoplasm [6,8,38]. Cytoplasmic GMPPB aggregates are degraded via the autophagy-lysosome pathway and accompanied by upregulation of microtubule-associated protein 1 light chain 3-II (LC3-II). The degradation process could be blocked and reversed by the lysosomal inhibitor leupeptin [37].

## 10. Genotype–Phenotype Correlation

Genotype–phenotype correlations in GMPPB-related disorder are difficult to confirm as most patients carry compound heterozygous mutations. Significant intrafamilial phenotype variability regarding the severity of weakness, the spectrum of central nervous system involvement, and ophthalmologic abnormalities have also been reported [27]. Nevertheless, the two most common mutations, c.79G>C (p.Asp27His) and c.860G>A (p.Arg287Gln), occur frequently enough to provide initial insight based on observations. Patients carrying the p.Asp27His mutation seem to have a milder LGMD phenotype, while patients with the p.Arg287Gln mutation frequently present with severe CMD affecting brain development [8,20,22]. Mutations reported to cause impairment of neuromuscular transmission can reside in any domain of the protein (Table 1). The mechanism leading to the broad spectrum of phenotype remains poorly understood. The milder clinical presentation of patients with p.Asp27His may be due to the mutation’s neutral effect on protein stability and normal cytoplasmic localization of mutant GMPPB. On the other hand, p.Arg287Gln has a stabilizing effect on the protein but disturbs its subcellular localization resulting in cytoplasmic GMPPB protein aggregation [8,20]. In addition, the presence of p.Arg287Gln may predict a severe phenotype when co-existing with a mutation leading to altered GMPPB protein stability (e.g., p.Ile219Thr or p.Pro32Leu). Conversely, p.Arg287Gln in compound heterozygosity with a neutral mutation (e.g., p.Gly220Arg or p.Val330Ile) seems to predict a less severe phenotype [20]. An adjacent mutation, c.859C>T, leading to a different substitution of Arg287 (p.Arg287Trp), has also been commonly reported but found more frequently in patients with LGMD phenotype. A study has suggested that the cellular level of GDP-mannose could correlate with the phenotype [30]. This study demonstrated markedly reduced enzymatic activity in all tested GMPPB mutants. Mutations in the catalytic domain of GMPPB were shown to affect its enzymatic activity more than mutations in other domains, leading to a more severe phenotype. This same study demonstrated that the enzymatic activity of GMPPB mutants could have a crucial role in the phenotypic severity of a zebrafish model of GMPPB disease.

## 11. Treatment

Patients with GMPPB-related disorder who have clinical and electrophysiologic features of neuromuscular transmission defect can respond to acetylcholinesterase inhibitors, such as pyridostigmine, alone or combined with 3,4 diaminopyridine or salbutamol. Improvement in strength (particularly in proximal muscles), endurance, and reduced use of gait aids, including wheelchairs, have been demonstrated in many patients [8,10]. The benefits of pyridostigmine, measured by the six-minute walking test and North Star Ambulatory Assessment, can steadily persist for 12 months. Its beneficial effect, however, decreases at 30 and 40 months, as detected by various scales, but the functional motor status of patients remains still significantly better compared to pre-treatment [35]. The associated progression of the dystrophic features could contribute to the progression of the disease and decrease the beneficial effects of the medications. It was suggested that the early introduction of pyridostigmine could protect from the development of scoliosis. A short course of corticosteroid therapy (0.75 mg/kg/day of prednisone for 3 months) in a single case of a 9-year-old boy with GMPPB-related CMD was reported to improve muscle strength, function scores, and CK levels [33].

Preliminary studies in a zebrafish model of GMPPB disease suggested that supplementation with GDP-mannose can rescue the phenotype, opening avenues to possible therapeutic strategies in GMPPB-related disorders, such as supplementation with GDP-mannose or similar molecules [30]. Additionally, as the decreased levels of certain GMPPB mutants (e.g., Ile268Thr) could be rescued by lysosomal inhibitors [30], drugs suppressing lysosomal activity could be potentially beneficial in ameliorating the phenotype of GMPPB-related disorders. Further studies are needed, however, to better explore novel therapeutic interventions.

## 12. Conclusions

Mutations in GMPPB affect the early stages of the glycosylation pathway and result in a broad spectrum of phenotypes ranging from CMD with brain and eye abnormalities to a mild form of LGMD, rhabdomyolysis, or CMS. CK levels are usually elevated. Like other dystroglycanopathies, patients typically show dystrophic changes and a patchy expression of α-DG on muscle biopsies. However, the muscle biopsy may be normal. Some patients demonstrate a defect of neuromuscular transmission, which classically spares facial muscles and can respond to cholinergic drugs. Such a defect of neuromuscular transmission and the β-DG electrophoretic mobility change are unique features of GMPPB-related disorders among dystroglycanopathies.

## Figures and Tables

**Figure 1 genes-14-00372-f001:**
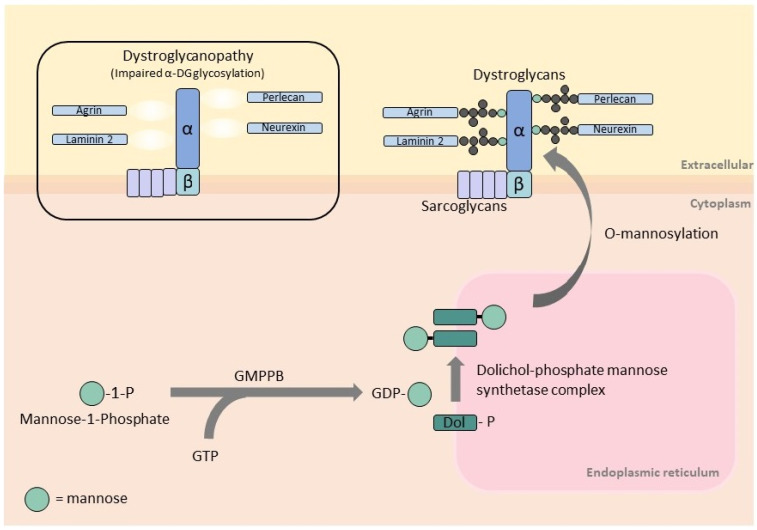
Schematic representation of GMPPB function in the O-mannosylation pathway and its association with α-dystroglycan. GMPPB catalyzes the formation of GDP-mannose from GTP and mannose-1-phosphate. GDP-mannose is the substrate of cytosolic mannosyltransferases required for the synthesis of dolichol-phosphate mannose, which is the essential mannose donor for O-mannosylation, a process required for the glycosylation of α-dystroglycan. Dystrophin–glycoprotein complex is linked to extracellular proteins (laminin 2, agrin, neurexin, perlecan, etc.) via glycans attached to α-dystroglycan. Impaired α-dystroglycan glycosylation disrupts the link to these extracellular proteins resulting in dystroglycanopathy/muscular dystrophies.

**Figure 2 genes-14-00372-f002:**
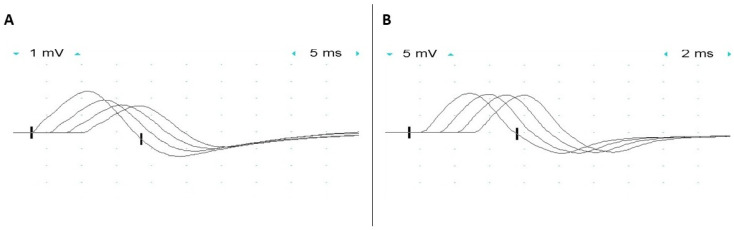
Electrophysiologic studies of a patient with GMPPB limb-girdle muscular dystrophy who had intermittent exacerbations of weakness and fatigable proximal muscle weakness. Low frequency (2 Hz) repetitive nerve stimulation demonstrated (**A**) a 35% decrement in compound muscle action potential amplitude in the trapezius upon stimulation of the spinal accessory nerve while (**B**) no decrement was observed in the abductor digiti minimi upon stimulation of the ulnar nerve. (Courtesy of Dr. Eric Sorenson and Ms. H. Ouelette).

**Figure 3 genes-14-00372-f003:**
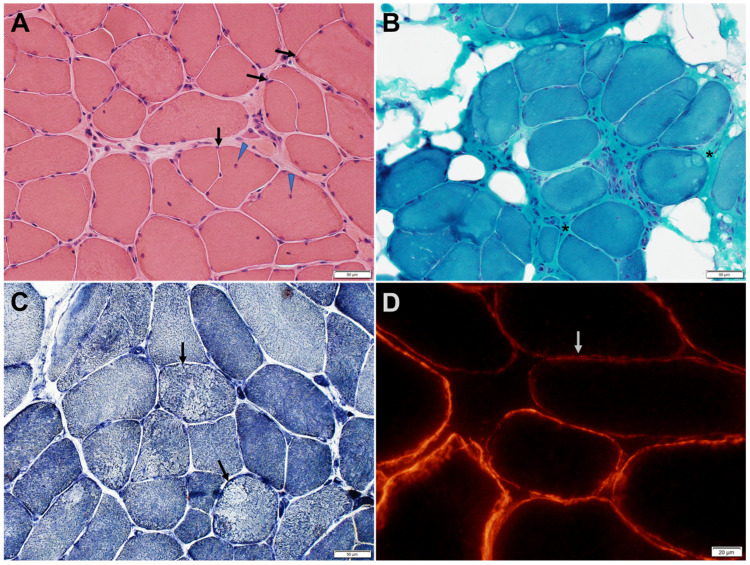
Muscle biopsy of a patient with GMPPB limb-girdle muscular dystrophy. (**A**) H&E-stained section shows variation in muscle fiber size, fiber splitting (arrows), and an increase in internalized nuclei (arrowheads). (**B**) Trichrome-stained section shows increased endomysial and perimysial fibrous and fatty connective tissue (asterisks). (**C**) Nicotinamide adenine dinucleotide dehydrogenase reacted section shows focal decrease of enzyme reactivity in a few fibers (arrows). (**D**) Immunohistochemical stain for α-dystroglycan shows patchy or diffuse reduction of α-dystroglycan immunoreactivity (arrow shows a representative fiber) at the sarcolemma of many muscle fibers.

**Figure 4 genes-14-00372-f004:**
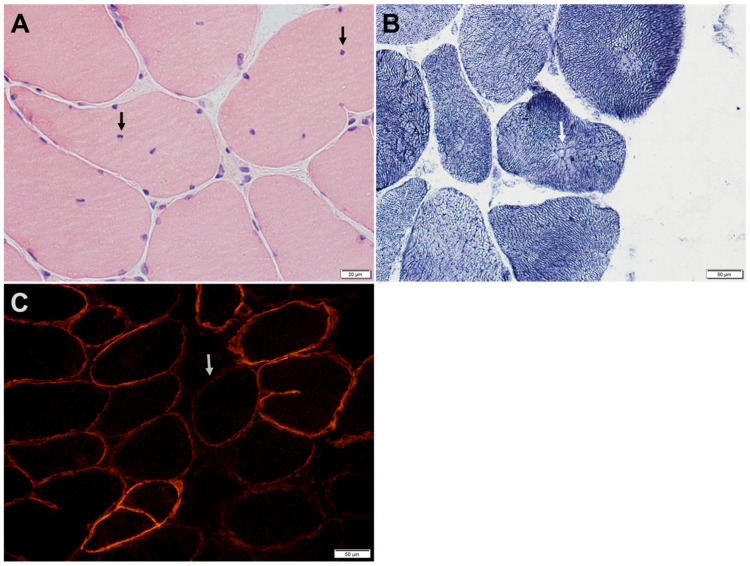
Intercostal muscle biopsy of a patient with GMPPB-related disorder manifesting with proximal weakness and defect of neuromuscular transmission. (**A**) H&E-stained section shows internalized nuclei (arrows) in several fibers. (**B**) Nicotinamide adenine dinucleotide dehydrogenase reacted sections shows central nucleation with radial arrangement of myofibrils (arrow). (**C**) Immunohistochemical stain for α-dystroglycan shows patchy reduction of α-dystroglycan immunoreactivity (arrow shows a representative fiber) at the sarcolemma of many muscle fibers.

**Table 1 genes-14-00372-t001:** Reported *GMPPB* mutations and associated phenotypes.

Domain	Mutation, DNA	Mutation, Protein	Reported Phenotypes	Reported Involvement of NMJ	Number of Patients/Families	References
**N-terminal nucleotidyl transferase domain (2-235aa)**	c.64C>T	p.Pro22Ser	CMD		2/2	[6,8]
	c.79G>C	p.Asp27His	LGMD	Yes	38/31	[6,7,8,10,20,21,22,23,24,25,26,27,28,29]
	c.87C>A	p.Cys29*	LGMD		1/1	[21]
	c.91A>G	p.Lys31Glu	CMD		1/1	[20]
	c.94C>T	p.Pro32Ser	CMD		1/1	[22]
	c.95C>T	p.Pro32Leu	LGMD, CMD		9/8	[6,20,21,24]
	c.130-3C>G	Splicing	LGMD	Yes	1/1	[8,10]
	c.220C>T	p.Arg74*	CMD		1/1	[6]
	c.308C>T	p.Pro103Leu	LGMD	Yes	3/1	[8,10]
	c.332T>G	p.Val111Gly	CMD		1/1	[30]
	c.338G>A	p.Cys113Tyr	LGMD		1/1	[21]
	c.346C>T	p.Pro116Ser	LGMD		1/1	[31]
	c.391G>T	p.Gly131Cys	CMD, LGMD		1/1	[32]
	c.395C>G	p.Ser132Cys	CMD		1/1	[22]
	c.402+1G>A	Splicing	LGMD	Yes	2/2	[22,29]
	c.458C>T	p.Thr153Ile	LGMD, CMD Recurrent rhabdomyolysis	Yes	5/4	[9,20,24,33]
	c.464G>A	p.Arg155His	LGMD		1/1	[25]
	c.503C>T	p.Ser168Phe	LGMD		1/1	[20]
	c.520G>A	p.Gly174Ser	CMD		1/1	[20]
	c.553C>T	p.Arg185Cys	CMD, LGMD	Yes	7/5	[6,31,34,35]
	c.559C>T	p.Gln187*	CMD	Yes	1/1	[8,10]
	c.578T>C	p.Ile193Thr	CMD	Yes	1/1	[8,10]
	c.640G>A	p.Gly214Ser	CMD		1/1	[30]
	c.656T>C	p.Ile219Thr	CMD (epilepsy)		6/5	[8,20,22,36]
	c.658G>C	p.Gly220Arg	LGMD		1/1	[20]
	c.700C>T	p.Gln234*	CMD, LGMD		4/3	[20,33]
**Inter-domain**	c.721C>T	p.Pro241Ser	LGMD, ID		1/1	[22]
	c.727C>T	p.Arg243Trp	LGMD		2/2	[20,21]
	c.754G>T	p.Gly252Cys	LGMD		1/1	[21]
	c.760G>A	p.Val254Met	LGMD	Yes	4/4	[8,10,22,25,29]
**LbH domain (259-294aa)**	c.781C>T	p.Arg261Cys	LGMD	Yes	1/1	[8,10]
	c.790C>T	p.Gln264*	LGMD, ID, epilepsy		3/1	[27]
	c.797G>A	p.Cys266Tyr	LGMD (respiratory involvement at 70)		2/1	[24]
	c.803T>C	p.Ile268Thr	LGMD	Yes	1/1	[37]
	c.810_813delCAATinsTGGC	p.Asn271Gly	CMD		4/3	[21]
	c.841G>A	p.Glu281Gln	LGMD		1/1	[21]
	c.854_855delGT	p.Cys285Tyrfs*19	LGMD		1/1	[20]
	c.859C>T	p.Arg287Trp	LGMD, Exercise intolerance, CMD	Yes	10/9	[8,10,20,21,22,25,26,28,29]
	c.860G>A	p.Arg287Gln	CMD (more common), LGMD	Yes	19/17	[6,8,21,22,23,24,29,36]
	c.863G>A	p.Arg288Gln	LGMD		1/1	[20]
	c.877C>T	p.Arg293Trp	Myalgia	Yes	1/1	[38]
**C-terminal**	c.902C>G	p.Ser301Cys	LGMD	Yes	1/1	[25]
	c.907C>T	p.Leu303Phe	LGMD	Yes	2/2	[8,10,21]
	c.943G>A	p.Gly315Ser	LGMD, Exercise intolerance, rhabdomyolysis, ID		1/1	[7,20]
	c.966C>A	p.Asn322Lys	LGMD	Yes	3/1	[38]
	c.988G>A	p.Val330Ile	LGMD, Elevated CK, Exercise intolerance		5/4	[6,20,21]
	c.1000G>A	p.Asp334Asn	CMD, LGMD	Yes	4/4	[6,8,21,39]
	c.1018G>A	p.Gly340Arg	Exercise intolerance		1/1	[38]
	c.1034T>C	p.Val345Ala	LGMD, ID		1/1	[22]
	c.1036C>A	p.Arg346Ser	LGMD, ID		3/2	[24,40]
	c.1039_1043dup	p.Ile349Serfs	LGMD	Yes	1/1	[25]
	c.1060G>A	p.Gly354Ser	LGMD	Yes	1/1	[37]
	c.1069G>A	p.Val357Ile	CMD, LGMD	Yes	5/5	[21,22,25,29]
	c.1070G>A	p.Arg357His	LGMD, Exercise intolerance	Yes	5/3	[38]
	c.1081G>A	p.Asp361Asn	CMD		1/1	[21]
	c.1090T>A	p.Tyr364Asn	LGMD		1/1	[32]
	c.1108G>C	p.Val370Leu	LGMD		1/1	[21]
	c.1151G>A	p.Arg384His	LGMD	Yes	1/1	[40]

**Abbreviations:** CMD, congenital muscular dystrophy; ID, intellectual disability; LbH, left-handed β-helix; LGMD, limb-girdle muscular dystrophy; NMJ, neuromuscular junction.

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
