# Peer review of "GDP-Mannose Pyrophosphorylase B (GMPPB)-Related Disorders"

_genes, 2023, doi:10.3390/genes14020372_

Round 1
Reviewer 1 Report
Hi,
I think this is good and comprehensive review of GMPPB-related disorders and those in the filed will find it useful.
I only have a few minor comments.
1. On Table 1, would you be able to insert a column and state the number of cases in which each mutation has been reported please? It would give useful information to be able to get a feel for just how common some of the mutations are. In the text certain mutations are stated as being the most common but no numbers are given.
2. I'm not clear about the sentence at lines 107-108 which states "Moreover, mutations affecting residues located between Pro241 and Asn271 are frequently observed in affected individuals". Why have these residues been specifically mentioned? Are they more common than other residues? Is there something special about the location in the protein? Could this be made a bit more clear please. Maybe addition of the column in table 1 that I've suggested above would help?
3. I know you've mentioned it later in the review, but in the section called Prevalence it might also be worth mentioning that NMJ involvement might be under-diagnosed if RNS is carried out in an unaffected muscle.
4. I'm afraid that my download seemed to miss out a lot of the alpha symbols preceding "-DG", so I'm not sure if the sentence that begins on line 217 is referring to alpha or beta DG: "- DG immunoreactivity is normal." If it is alpha, then isn't this sentence just basically repeating the previous sentence?
5. First word on line 241 is pattern I think?
Author Response
REPLY TO REVIEWER 1
We thank very much Reviewer 1 for the suggestions and questions, which we have addressed, improving the manuscript.
- On Table 1, would you be able to insert a column and state the number of cases in which each mutation has been reported please? It would give useful information to be able to get a feel for just how common some of the mutations are. In the text certain mutations are stated as being the most common but no numbers are given.
REPLY: A column showing number of patients/families with each mutation was added to Table 1.
- I'm not clear about the sentence at lines 107-108 which states "Moreover, mutations affecting residues located between Pro241 and Asn271 are frequently observed in affected individuals". Why have these residues been specifically mentioned? Are they more common than other residues? Is there something special about the location in the protein? Could this be made a bit more clear please. Maybe addition of the column in table 1 that I've suggested above would help?
REPLY: The statement was based on observations in a large cohort study. However, as we have now added the additional column in Table 1 showing frequencies of each mutation as suggested above, the sentence about Pro241 and Asn271 was removed from the text.
- I know you've mentioned it later in the review, but in the section called Prevalence it might also be worth mentioning that NMJ involvement might be under-diagnosed if RNS is carried out in an unaffected muscle.
REPLY: We added the sentence “Moreover, the defect of neuromuscular transmission may be undetected if repetitive nerve stimulation is performed in an unaffected facial or distal muscle” in the Prevalence section.
- I'm afraid that my download seemed to miss out a lot of the alpha symbols preceding "-DG", so I'm not sure if the sentence that begins on line 217 is referring to alpha or beta DG: "- DG immunoreactivity is normal." If it is alpha, then isn't this sentence just basically repeating the previous sentence?
REPLY: The questioned sentence describes beta-DG immunoreactivity while the previous sentence refers to alpha-DG immunoreactivity for different epitopes. For clarity, we slightly modified the first sentence, and the text now states the following: “Muscle biopsy of patients with GMPPB-related disorders, immunostained with antibodies directed against the IIH6 epitope, show variable reduction of alpha-DG expression, while a-DG expression appears normal when muscle is immunostained with antibody directed against the core protein. Beta-DG immunoreactivity is normal”.
- First word on line 241 is pattern I think?
REPLY: Apology. The typo was corrected.
Reviewer 2 Report
The authors provide a comprehensive and well-written review of GMPPB-dystroglycanopathies with a good literature review.
Here are some suggestion that may help to improve the structure of the paper:
- purpose of the review could be added to the Introduction
- Is there other evidence of clues (clinical picture, imaging, electrophysiological studies..), other than electrophoretic mobility changes, that can help physicians distinguish GMPPB from other secondary dystroglycanopathies?
- The clinical spectrum description is a bit confusing, as LGMD phenotype is described along the CMD without a clear definition of none of them and CMD phenotype is not uniformly described along the subchapter.
- can authors deeper discuss the etiopathogenetic mechanism of NMJ involvement (CMS symptoms)?
- About genotype-phenotype correlation: is there evidence supporting it? it is known that, for example, no correlation exists between the clinical severity and the extent of hyposyalisation…
Finally, which immunostain is shown in Fig 3 and 4?
Author Response
REPLY TO REVIEWER 2
We thank very much Reviewer 2 for the suggestions and questions, which we have addressed, improving the manuscript.
purpose of the review could be added to the Introduction
REPLY: The purpose of the review was added to the end of the Introduction.
Is there other evidence of clues (clinical picture, imaging, electrophysiological studies..), other than electrophoretic mobility changes, that can help physicians distinguish GMPPB from other secondary dystroglycanopathies?
REPLY: Clinical and electrophysiologic evidence of neuromuscular junction defect and β-DG electrophoretic mobility change are unique features that help distinguishing GMPPB-related disorders from other dystroglycanopathies. This is summarized in Conclusion.
The clinical spectrum description is a bit confusing, as LGMD phenotype is described along the CMD without a clear definition of none of them and CMD phenotype is not uniformly described along the subchapter.
REPLY: The section Clinical Phenotype section was rewritten, as suggested. Clinical spectrum is now subdivided into LGMD, CMD, and CMS.
can authors deeper discuss the etiopathogenetic mechanism of NMJ involvement (CMS symptoms)?
REPLY: Pathomechanism of the post-synaptic NMJ impairment has been added to the section Electrophysiologic Findings. The possible pathomechanism underlying the presynaptic dysfunction, due to impaired agrin glycosylation, had been already described in the same section.
About genotype-phenotype correlation: is there evidence supporting it? it is known that, for example, no correlation exists between the clinical severity and the extent of hyposyalisation…
REPLY: The genotype-phenotype correlation of the two most common mutations is mainly based on observations. In the section Genotype-Phenotype we added: 1) Additional observation on a specific mutation (p.Arg287Trp); 2) Role of GMPPB enzymatic activity on phenotype.
Effects of mutations on protein stability, localization, and aggregation, potentially contributing to phenotypes, had been already discussed.
Finally, which immunostain is shown in Fig 3 and 4?
REPLY: We apologize for the lack of clarity. Figures show immunostain for α-dystroglycan. This information was added to figure legends.
Others: We corrected occasional typos we noted in the text.